# Investigation of Molecular Weight, Polymer Concentration and Process Parameters Factors on the Sustained Release of the Anti-Multiple-Sclerosis Agent Teriflunomide from Poly(*ε*-caprolactone) Electrospun Nanofibrous Matrices

**DOI:** 10.3390/pharmaceutics14081693

**Published:** 2022-08-14

**Authors:** Nikolaos D. Bikiaris, Ioanna Koumentakou, Georgia Michailidou, Margaritis Kostoglou, Marilena Vlachou, Panagiotis Barmpalexis, Evangelos Karavas, George Z. Papageorgiou

**Affiliations:** 1Laboratory of Polymer Chemistry and Technology, Department of Chemistry, Aristotle University of Thessaloniki, 54124 Thessaloniki, Greece; 2Laboratory of General and Inorganic Chemical Technology, Department of Chemistry, Aristotle University of Thessaloniki, 54124 Thessaloniki, Greece; 3Section of Pharmaceutical Technology, Department of Pharmacy, School of Health Sciences, National and Kapodistrian University of Athens, Panepistimioupoli-Zografou, 15784 Athens, Greece; 4Department of Pharmaceutical Technology, School of Pharmacy, Aristotle University of Thessaloniki, 54124 Thessaloniki, Greece; 5Pharmathen S.A., Pharmaceutical Industry, Dervenakion Str. 6, Pallini Attikis, 15351 Athens, Greece; 6Department of Chemistry, University of Ioannina, P.O. Box 1186, 45110 Ioannina, Greece

**Keywords:** electrospinning, poly(ε-caprolactone) PCL, nanofibers, process parameters, teriflunomide, drug release

## Abstract

In the current work, a series of PCL polyesters with different molecular weights was synthesized and used for the fabrication of nanofibrous patches via electrospinning, as sustained release matrices for leflunomide’s active metabolite, teriflunomide (TFL). The electrospinning conditions for each sample were optimized and it was found that only one material with high Mn (71,000) was able to produce structures with distinct fibers devoid of the presence of beads. The successful preparation of the fibers was determined by scanning electron microscopy (SEM).TFL (10, 20 and 30 wt%) in three different concentrations was incorporated into the prepared nanofibers, which were used in in vitro drug release experiments. The drug-loaded nanofibrous formulations were further characterized by Fourier transform infrared spectroscopy (FTIR), differential scanning calorimetry (DSC) and powder X-ray diffractometry (XRD).It was found that TFL was incorporated in an amorphous form inside the polymeric nanofibers and that significant molecular interactions were formed between the drug and the polyester. Additionally, in vitro dissolution studies showed that the PCL/TFL-loaded nanofibers exhibit a biphasic release profile, having an initial burst release phase, followed by a sustained release until 250 h. Finally, a kinetic analysis of the obtained profiles revealed that the drug release was directly dependent on the amount TFL incorporated into the nanofibers.

## 1. Introduction

Teriflunomide (TFL) is a pyrimidine synthesis inhibitor, used for the treatment of multiple sclerosis (MS), which has demonstrated clinical efficacy and safety in a number of large and multicenter clinical trials. Recently, TFL received approval by both the European Medicines Agency (EMA) and the US Food and Drug Administration (FDA) as an oral administrated disease-modifying therapy (DMT) for the treatment of relapsing–remitting MS (RRMS) in adults [1]. Generally, the oral administration of active pharmaceutical ingredients (APIs) is amongst the most easily handled and economical routes for drug delivery. Nonetheless, TFL’s undesirable side-effects, such as high toxic levels and untargeted cells detection, and the risk of hepatoxicity and teratogenicity of TFL have led researchers to develop alternatives routes of administration [2]. 

Transdermal drug delivery systems (TDDS) have been exploited as a successful sustained drug release platform and have received regulatory approval for a series of products, such as patches. This type of drug delivery also allows for less frequent dosing or steady delivery profiles and may be easily self-applied with a painless and noninvasive application [3]. In most cases, sustained-release patches exhibit a high valuable profile since the delivery of the active biomolecules can be prolonged and controlled by a diffusion mechanism [4]. Owing to their interesting properties, such as a high surface area, high drug loading capacity and porosity, electrospun fibers from biodegradable and biocompatible polymers have recently received significant attention in the biomedical field [5,6]. Lately, there has been a great number of research studies that have explored the success of electrospun fibers from biodegradable polymers as vehicles for TDDS. Indicatively, Siafaka et al. fabricated poly(lactic acid)/poly(butylene adipate) (PLA/PBAd) electrospun blends for the controlled release of teriflunomide [4]. Sa’adon et al. reviewed the factors affecting the transdermal drug release from poly(vinyl alcohol) (PVA) electrospun nanofibers [7] and Ravikumar et al. prepared poly(caprolactone)-co-poly(ethylene glycol) (PCL-PEG) electrospun nanofibrous patch for the transdermal delivery of tetrahydro curcumin [8]. The major advantages of drug-loaded fiber mats include biocompatibility and biodegradability, so as to control the drug release and burst effect ensuring the desirable long-term delivery or immediate action at the targeted location [9].

There is an enormous flexibility in materials used for the preparation of electrospun nanofibers in drug delivery systems for various drugs. Polycaprolactone (PCL) is a US-FDA-approved, biocompatible and non-immunogenic semicrystalline polymer which exhibits a slow biodegradation rate [8,10,11,12,13,14]. The combination of the PCL properties (biocompatibility and slow biodegradation), such as high mechanical properties and the unique aspect of nanofibrous structure originating from electrospinning can result in a promising material for medical applications including TDDS [15,16]. 

In this context, we used electrospinning technique to produce PCL fibers loaded with teriflunomide, for the treatment of multiple sclerosis. Relevant factors influencing the fiber formation and the mean fiber diameter, such as molecular weight and polymer concentration [17], flow rate, electric field and distance between needle and the collector [18] were thoroughly examined. Any of these parameters may have a direct effect on the morphology and diameter size of the fibers. Moreover, a careful adjustment of these parameters can result in fibers with the desired morphology and diameter size [19]. The aim of the current work was to synthesize PCL polyesters, determine the influence of the molecular weight of polycaprolactone, solvents and operational conditions on the diameter of nanofibers during the electrospinning process and study the loading of TFL into the prepared nanofibrous networks.

## 2. Materials and Methods

High Purity TFL (Figure 1) was kindly donated from Pharmathen S.A (Athens, Greece). ε-Caprolactone (ε-CL) monomer of 99% purity was purchased from Alfa Aesar (Kandel, Germany) and was further purified by distillation (2×). Tin(II) 2-ethylhexanoate (TEH) catalyst of 92.5–100% purity was purchased from Sigma-Aldrich chemical company (Saint Louis, MO, USA). Toluene of 99.8% purity was purchased from Nv Chem-lab (Zegeldem, Belgium) and used as received. 1.4-butanediol acting as coinitiator of 99% purity was purchased from Alfa Aesar (Kandel, Germany) and used as received. All other reagents used were of analytical grade.

### 2.1. Synthesis of the PCL Polyesters

The three PCL polyesters were prepared by the widely employed the ring-opening polymerization (ROP) of the *ε*-caprolactone method as reported in a previous work of our group [20]. The bulk polymerization of ε-CL was carried out in a 250mL round-bottom flask equipped with a mechanical stirrer under vacuum. A solution of the TEH catalyst in toluene (50 mg/mL), acting as initiator (I), was added (final concentration 400 ppm), followed by 1,4-butanediol, acting as co-initiator (CI), at different concentration ratios with respect to the monomer (Table 1). The polymerization mixture was degassed and purged with dry nitrogen three times. The ROP reaction was carried out for 2 h at 190 °C, and the temperature was then raised to 210–240 °C over a period of 45 min. Unreacted monomer was removed slowly by high vacuum (5 Pa) distillation, to avoid excessive foaming, over a time period of 15 min. The polymerization was terminated by rapid cooling of the flask to room temperature. The PCL samples obtained had molar masses Mn ~13, ~24 and ~71 kg/mol, as was determined by size-exclusion chromatography (SEC).

### 2.2. Preparation of Electrospun Nanofibers

A standard electrospinning setup (FLUIDNATEK^®^ LE-10, Bioinicia, Valencia, Spain), equipped with a 10 cm diameter rotating drum collector) was used for the fabrication of the electrospun PCL nanofibers (Figure 2). Three different % *w/v* concentrations (12, 15 and 17% *w/v*) of PCL in chloroform were prepared. Different set of parameters (Table 2) were examined, in order to obtain continuous fibers with smooth surfaces, devoid of the presence of any beads and other defects. PCL-based electrospun patches were fabricated at an applied voltage ranging from 12 kV to 17 kV, using a high-voltage power supply. The ground collection plate of aluminum foil was located at a fixed distance of 10 and 12 cm from the needle tip. The speed of the rotated collector was set at 1000 rpm and the feeding rate to 500 and 1000 μL/h, and each sample was subjected to the electrospinning process at 25.0 ± 0.1 °C for 30 min. The collected electrospun samples were dried for 24 h at ambient temperature to remove chloroform. The electrospinning experiments were performed at room humidity and room temperature.

### 2.3. Incorporation of TFL into the Electrospun Fibers

Teriflunomide (10, 20 and 30 wt%) was dissolved in a 17% (*w/v*) PCL/chloroform solution. An SEM morphology analysis of the investigated parameter systems showed that set 7 was the optimal one, in terms of mean fibers diameter. The electrospinning process was assessed at room temperature 25.0 ± 0.1 °C and each sample was left for 150 min. The collected electrospun samples were dried for 24 h, at ambient temperature to remove chloroform. The electrospinning experiments were performed at room humidity and room temperature.

### 2.4. Characterization Techniques

#### 2.4.1. Size-Exclusion Chromatography (SEC)

Molar mass (Mn, Mw) determination was performed using size-exclusion chromatography (SEC). The measurements were performed on the samples as received upon their preparation. The analysis was performed in an SEC equipment consisting of a Waters 600 high-pressure liquid chromatography pump (Waters, Milford, MA 01757, USA), Waters Ultrastyragel columns (HR-1, HR-2, HR-4, HR-5) and a Shimadzu RID-10A refractive index detector (Shimadzu Corporation, Kyoto, Japan). Column calibration was performed using polystyrene standards (1–300 kDa/mol in molecular weight). The concentration of the prepared solutions was 20 mg/1000 μL, the injection volume was 150 μL and the flow rate 1 mL/min.

#### 2.4.2. Scanning Electron Microscopy and Micrographs Analysis

The surface morphology characterization of the fabricated electrospun PCL nanofibrous patches was performed using field-emission scanning electron microscopy, JEOL JSM-7610F Plus, supported by an Oxford AZTEC ENERGY ADVANCED X-act energy dispersive X-ray spectroscopy (EDS) system (JEOL Ltd., Tokyo, Japan). A 200 Å thick carbon coating was applied to increase the conductivity of the samples. The morphometric analysis and mean diameter size of the electrospun nanofibers was quantitatively performed by processing SEM micrographs using ImageJ software, an image processing program designed for scientific multidimensional images. Ten different segments on each image were randomly measured to obtain an average fibers diameter. The fiber morphology was visually evaluated in terms of uniformity, smoothness, absence of beads and defections. 

#### 2.4.3. Fourier Transform Infrared Spectroscopy (FT-IR)

FT-IR spectra of samples were obtained using an FT-IR spectrometer (model FTIR 2000, Perkin Elmer, Waltham, MA, USA). A small amount of each sample was triturated with the proper amount of potassium bromide (KBr), and disks were formed under pressure. The spectra were collected in the range of 400 to 4000 cm^−1^, at a resolution of 4 cm^−1^, using 64 coadded scans, and the baseline was corrected and converted into absorbance mode.

#### 2.4.4. X-ray Diffraction (XRD)

X-ray diffraction (XRD) patterns were performed with an XRD-diffractometer (Rigaku Miniflex II, Beijing, China) with a CuKα radiation for crystalline phase identification (λ = 0.15405 nm for CuKα). The sample was scanned from 5 to 60, with steps of 0.05°.

#### 2.4.5. Differential Scanning Calorimetry (DSC)

For the DSC analysis, a PerkinElmer Pyris 1 differential scanning calorimeter (Waltham, MA, USA), calibrated with indium and zinc standards, was used. About 5.0 ± 0.1 mg of each sample was weighed, placed in sealed aluminum pans and heated up from 20 to 240 °C, with a heating rate of 20°C/min in an inert atmosphere (N_2_, flow rate 50 mL/min), held in for 3 min, in order to erase any thermal history, cooled to approximately 20 °C with a cooling rate of 20 °C/min and heated again up to 240 °C, with a heating rate of 10 °C/min.

### 2.5. Drug Loading and Drug Content Quantification 

After the electrospinning process, preweighed samples (10 mg) were cut from PCL_TFL10%, PCL_TFL20% and PCL_TFL30% samples and dissolved in 50 mL of methanol (MeOH) and the resultant solution was analyzed for TFL content, using high-performance liquid chromatography (HPLC). The column used was a CNW Technologies Athena C18, 120 A, 5 μm, 250 mm × 4.6 mm (column temperature 25 °C). The mobile phase consisted of acetonitrile and ultrapure water ACN/H_2_O (acidified with acetic acid at final pH = 3) 80/20 *v*/*v*, at a flow rate of 1.0 mL/min). The concentration was measured by using a HPLC–UV apparatus at 295 nm and was based on a previously created calibration curve. The dilution time was 12 min, while the injection volume was set at 20 μL. The calibration curve was created by diluting a stock methanol solution of 1 mg/mL TFL to concentrations of 0.01, 00.2, 0.05, 0.25, 0.5, 0.75, 1, 2.5, 5, 10, 20 and 50 ppm using the mobile phase.

Drug loading (DL) was calculated according to the following equation:(1)$$\mathrm{Drug}\;\mathrm{Loading}\;(\mathrm{DL})=\frac{W_{\mathrm{experimental}}}{W_{\mathrm{theoretical}}}\;\times \;100\%$$

### 2.6. In Vitro Drug Release

The TFL’s in vitro release, was studied by using a DISTEK Dissolution Apparatus Evolution 4300, equipped with an autosampler using the paddle method (USP II method). A film was placed on each dissolution vessel corresponding to approximately 10 mg of each formulation in an appropriate transdermal patch holder, with its application side up. The test was performed at 37 ± 0.5 °C with a rotation speed of 100 rpm for 132 h. The volume of the phosphate-buffered saline (PBS), pH = 7.4, dissolution medium was 400 mL. Two (2) milliliters of the aqueous solution was withdrawn from the release media and analyzed for the TFL content with the aid of the HPLC method, using the same conditions as described in 2.5.

## 3. Results

### 3.1. Characterization of Newly Synthesized PCL Polyesters

The newly synthesized PCL polyesters were fully characterized by Klonos et al. in a previous work [20]. Briefly, these biobased polyesters were studied combining a sum of experimental techniques for the structure (FT-IR spectroscopy, X-ray diffraction), thermal transitions (calorimetry), nuclear magnetic spectroscopy (^1^H and ^13^C NMR) (Appendix A), semicrystalline morphology (polarized microscopy) and molecular dynamics (broadband dielectric spectroscopy).

### 3.2. Effect of Various Parameters on the Morphology of the Fibers

#### 3.2.1. Effect of the Molecular Weight

The molecular weight of the polymer has a great impact on its rheological behavior and electrical properties such as viscosity, surface tension, conductivity and dielectric strength [21]. In view of this, in the present work, three PCL polyesters with different number average molecular weights, Mn, of 13,400, 24,100 and 71,000 g/mol were prepared and investigated (Appendix A). Each of the polymers was dissolved in pure chloroform to define the minimum molecular weight necessary for the fabrication of uniform electrospun fibers devoid of beads. Table 3 presents the SEM micrographs of the electrospun PCL polyesters in chloroform solutions with different molecular weights. It is noteworthy that as the molecular weight increases, several variations can be spotted in the electrospun structure. 

#### 3.2.2. Effect of Polymer Concentration

Each of the prepared PCLs, with different molecular weights, was dissolved in three different concentrations in a chloroform solution (12, 15 and 17% *w/v*). The solvent system selected to dissolve the polymer is highly significant for the whole procedure. In the case of PCL fibrous mats, it was established that the use of volatile solvents such as chloroform is favorable in order to prepare uniform fibrous structures. Table 4 presents SEM images and the mean diameters of fibrous structures obtained for 71,000 g/mol in the above concentrations. The effect of the solution concentration on the structure is easily noticed. The average diameter size of the fibers and the interfiber spacing increases significantly as the concentration of the solution increases. Table 5 provides an overall qualitative evaluation and optical observations of the electrospun structures of all samples studied. To sum up, only PCL with 71,000 g/mol Mn was able to produce fibers, while the increase of the polymer concentration in the chloroform solution increased the interfiber spacing. 

#### 3.2.3. Effect of Voltage of the Electric Field

In the current study, PCL/chloroform solutions were electrospun under three different levels of applied voltage (12, 15 and 17 V) to produce a single jet without clogging or splitting. The effect of voltage of the electric field was investigated in PCL with Mn 71,000 in 17% *w/v* chloroform solution. It has been reported that the diameter of electrospun fibers is not significantly affected by the applied voltage of the electric field [22]. Figure 3 presents the effect of the applied voltage on the fiber diameter of the electrospinning mats. It was observed that the diameter of the electrospun fibers was not significantly altered by varying the applied voltage. A higher fiber diameter was obtained for the minimum value of applied voltage (12 V), while diameter values for 15 and 17 V were almost the same. 

#### 3.2.4. Effect of Flow Rate 

A similar trend was observed for the effect of volume feed rate on the diameter of electrospun fibers. For the same polymer solution under investigation as above, Figure 4 reveals that higher-diameter fibers were obtained when using a 500 μL/h flow rate with a non consistent effect, while a 1000 μL/h flow rate fabricated lower-diameter fibers with a consistent effect on fiber diameter. Usually, a higher flow rate results in thicker fibers due to the higher mass flow, and, above a critical value, the formation of beaded fibers may occur due to unstable jet formation.

#### 3.2.5. Effect of the Distance between Needle and Collector

Two different distances between the tip of the needle and the collector (10 and 12 cm) were investigated in the present work. Figure 5 exhibits the obtained results: it is evident that the distance in this work does not play a significant role in the obtained fibers diameter since similar size mats were obtained with the same consistency in both studied distances. A higher distance could lead to the thinning of the electrospun fibers since there will be a greater stretching distance. It has also been found in previous studies [23,24], and several interactions have been observed between applied voltage and spinning distance. 

### 3.3. Characterization of the Drug-Loaded Fibrous Mats with DSC, XRD, FTIR and SEM

The morphology of the fibers with different drug loadings is presented in Table 6. Scanning electron microscopy micrographs revealed that the nanofibers loaded with TFL were randomly aligned, beadless, interconnected and continuous. SEM coupled with an ImageJ analysis was conducted in order to calculate the mean fiber diameter of the fiber mats. The TFL content had an influence on the average diameter of PCL/teriflunomide fibers. It was seen that the diameter of the fibers (Table 7) was increased, regarding the neat PCL fibers, as expected due to the incorporation of different contents of teriflunomide. As the teriflunomide content increased to 30%, the average diameter of the drug-loaded fibers increased by as much as 0.653 μm (from 0.492 μm). The average diameter of PCL fibers containing 20% of drug was 0.541 μm.

Pure TFL exhibits characteristic peaks at 3305 cm^−1^ and 3067 cm^−1^, which correspond to strong absorptions of hydroxyl (-OH) and secondary amino groups (-NH), respectively, at 1636 cm^−1^ attributed to the carbonyl group (C=O) , and at 2220 cm^−1^, corresponding to the triple nitrogen carbon bond trend in the nitrile (−C≡N) group [25]. The most characteristic peak in PCL is recorded at 1744 cm^−1^ and is due to the absorption of the carbonyl group of the ester bonds. Thus, as seen from Figure 6, the IR spectrum of the neat PCL and neat TFL showed all said characteristic IR peaks. Looking at the obtained FT-IR spectra of the TFL-loaded nanofibers, it is obvious that when TFL was incorporated into the PCL polyester fibers, the absorption peak of the carbonyl group of PCL at 1636 cm^−1^ was shifted to 1633 cm^−1^ for PCL_TFL10%, 1632 cm^−1^ for PCL_TFL20% and 1634 cm^−1^ for PCL_TFL30%, respectively, probably due to the formation of significant intermolecular hydrogen bonds between the C=O of PCL and -NH of TRL. Furthermore, an additional small absorption peak shift was also observed from 2220 cm^−1^ to 2222–2225 cm^−1^ in all three samples, probably due to nitrile alkyl interactions [25]. 

Another important feature to be studied is the physical state of the active substance within the formulation, as well as its dispersion within the carrier. Usually, the amorphous dispersion is what helps enhance the solubility in low-solubility drug molecules. In that sense, the DSC and XRD methods were employed in order to investigate the physical state of TRL in the prepared fibrous mats. As it can be seen from the thermograms in Figure 7, PCL is a semicrystalline polymer with a melting point at 60 °C, which is in good agreement with that found in the literature [20]. Pure TRL is a crystalline drug with a melting point of 228.7 °C. However, as seen in the thermograms of the fibrous patches, this peak does not appear and only the T_m_ of PCL is recorded at temperatures similar to those found in the pure fibrous structures, without drug incorporation. 

TRL exhibits high crystallinity, like most drug molecules [25]. Figure 8 shows the diffractograms of pure TRL and of the prepared drug carriers. It is observed that the TFL shows characteristic peaks at 2θ of 15.1, 17.4, 18.3, 19.3, 21.2, 22.3, 24.3, 27.1 and 30.7°, which are all in accordance with the ones found in literature [26]. XRD confirms that the incorporation of TFL had a significant effect in the crystallinity of electrospun fibrous mats. The XRD measurement reveals three well-resolved peaks observed at 2θ of 20.6°, 21.2° and 23° indexed to the main peak at the (110), (111) and (200) lattice planes, respectively, of an orthorhombic crystalline structure of PCL [27]. Additional characteristic peaks corresponding to TFL are not noticed in the drug-loaded fibrous mats. This is attributed to the complete solubilization and amorphization of the drug within the polymeric matrix and is in accordance with the DSC thermographs analyzed previously.

### 3.4. In Vitro Dissolution Study of TFL from Electrospun Mats

From Figure 9, based on the release profile of pure TFL, it becomes apparent that after almost 50 h, a complete release has been occurred. However, when the drug is incorporated into the nanofibers it is noticed that its release rate is significantly decreased, compared to neat TFL. Looking more closely to the release profiles of TFL from the fibrous PCL mats, it is concluded that the release of TFL is controlled by a two-step release procedure. Specifically, a burst release phase is initially seen up to 10 h, which is followed by a sustained-controlled release phase up to 250 h. Apart from the initial burst release, the rate of TRL release from the PCL fibers is maintained nearly constant, suggesting a well-sustained release pattern. Additionally, it is observed in PCL fibers containing the highest content of TFL (PCL_TFL30%), that the active substance dissolves completely within 250 h, while in fibers with 10 wt% TFL, the lowest release rate is observed, reaching up to 60% in 240 h. Furthermore, intermediate release rates are seen for PCL fibers with 20 wt% TFL, where the release of TRL at 240h reaches almost 85% [28,29,30]. 

### 3.5. Investigation of TFL’s Release Mechanism

The morphological changes taking place in the membrane during the in vitro release are shown in Table 8. From the SEM microphotographs of PCL/TFL after the in vitro drug release, it is clear that there is a significant change in fibers morphology that takes place during release. As can be easily seen, the fibrous structures have been destroyed by the scission of the backbone ester groups from the buffer penetration and as a result, droplet-shaped particles are formed in their place. Moreover, the surface of the fibers became rough due to the surface erosion of the PCL matrix. This finding denotes that the degradation process starts from the surface of the PCL fibers and proceeds to the inner regions of the particles [31]. It is also evident that, as the contact of the TFL increases in the samples, so does the progression of the matrix erosion. A significant erosion of the particles is observed in the highest drug-loaded sample (PCL/TFL_30%), which is expected since a complete drug release at 250 h has occurred (Figure 9). 

### 3.6. Release Data Analysis

In this section, the modelling of the drug release data, presented above, is illustrated, starting from the neat TFL samples. Let us start from the neat TFL samples. The neat drug release is a dissolution process determined kinetically by two steps. The first kinetic step is the transition of the drug from the solid to the liquid phase. The second kinetic step is the mass transfer of the drug in liquid phase from the solid–liquid interface region to the bulk liquid region. Taking into account that (according to the release data) the characteristic release time is more than 1 h and that the system is agitated to enhance mass transfer, one can safely assume that the rate-determining step of the dissolution process is the transition of TFL from the solid to the liquid phase. This step can be represented as an apparent reaction. The order of this reaction for most drugs is one. This order corresponds to the well-known Noyes and Whitney equation [32]. However, in the present case this approach could not be represented by the experimental data. A general nth order apparent reaction rate was considered. The integrated kinetic equation for the evolution of released fraction X was
(2)$$X=1\;-\;(1\;+\;\left(n\;-\;1\right){\mathrm{Kt}}^{1/(1\;-\;n)}$$
where K is a kinetic constant and n is the reaction order. The percentage release was simply 100X so the above equation was fitted (through least squares minimization) to the experimental data of the neat TFL release. The fitting quality appears in Figure 10. The parameter values resulted from the fitting procedure are n = 1.78 and K = 1.48 × 10^−4^ s^−1^.

A further simplification step was to consider the approximate exponential form of solution of the diffusion equation [33]. Summarizing the above assumption, it was found that the release evolution was represented by the following equation:(3)$$X\;{=\;\varphi }_{1}\left(1-\exp\left({-k}_{1}t\right)\right){+\varphi }_{2}(1\;-\exp\left({-k}_{2}t\right)$$
where φ_1_ andφ_2_ are the fraction of the drug in the interior and exterior of the fibers, respectively, and k_1_ and k_2_ are the rate constants of the two diffusion mechanisms. The above equation was fitted to the percentage release data (equivalent to 100X) of Figure 9 by employing a least squares minimization method. The comparison between the model results and experimental data is shown in Figure 10. The R^2^ correlation coefficient was 0.97, 0.98 and 0.99 for 10%, 20% and 30% of drug content in the patches, respectively. The fitting parameters are shown in Table 9.

The coefficients k_1_ are two orders of magnitude larger than k_2_, confirming the assumption of individual evolution of the two diffusion processes. A clear trend cannot be deciphered for the dependence of the parameters on drug content from the expected increase of φ_1_. Using asymptotic formulae for the solution of the diffusion equation, it can be shown that the parameter k can be related to the diffusion coefficient D through a relation of the form k = αD/L^2^ [34], where both the coefficient α and the characteristic length L depend on the diffusion domain geometry. In the case of mechanism (i), the length L is the slab thickness and α = 3 [35]. In the case of a sphere, L is the radius of the sphere and α = 15 [36]. In the case of the mechanism (ii) where the geometry is evolved from cylindrical to spherical, L is the fiber radius and α is approximately taken as 10. Based on the above relation, assuming a patch thickness of 1 mm and employing the measured average fiber radius, the order of magnitude estimation for the diffusivities for mechanisms (i) and (ii) were10^−10^ m^2^/s and 10^−22^ m^2^/s. These are quite reasonable estimates for the diffusion of TFL in interfiber voids and intrafiber pores, respectively. 

## 4. Discussion

In our current research, poly(caprolactone) (PCL) polyesters with various molecular weights were synthesized and used for the fabrication of nanofibers with optimum morphology using the process of electrospinning for the preparation of delivery patches of teriflunomide, a drug used for the treatment of multiple sclerosis.

The electrospinning technique has received significant attention in the last years as an effective tool for the production of nano- to microscale polymeric fibers for various applications. Factors such as the interconnection of the pores, the high porosity as well as the ratio of high surface to the total volume of the fibrous patches make them particularly promising drug delivery systems.Electrospinning involves many scientific aspects, including polymer science, applied physics, fluid mechanics, electrical, mechanical, chemical, and material engineering, rheology and many others. Therefore, various parameters, divided into two main groups, regarding the solution properties and processing conditions can affect the shape and surface morphology of the fibers. The first group mainly includes molecular weight, solution viscosity and polymer concentration, while the second involves the applied voltage of the electric field, volume feed rate as well as the distance of the needle from the collector [37,38]. The optimization of these parameters could lead to the production of longer continuous fibers with the desired length and diameter [22]. For this reason, an extensive investigation regarding the molecular weight of PCL, polymer concentration in chloroform, applied voltage of the electric feed, volume feed and the distance of the needle from the collector was conducted.

The molecular weight of the polymer is considered as one the most significant aspects regarding the production of nanofibers devoid of beads, since it greatly affects the rheological properties of the electrospun solution, such as viscosity and surface tension. It was observed that the formation of fibrous structures at 13,400 and 24,100 Mn was not successful. Instead, bead structures were obtained, probably due to the resistance of the jet to the tensile flow. In addition, at 24,100 g/mol Mn, the number of beads was higher, and the spacing between beads was smaller compared to 13,400 g/mol. Thus, it seems that the molecular weight of the polymer has a significant effect on the formation of fibrous structures, since it can affect the breakup of the viscoelastic jet. It has been well established that in non-Newtonian fluids, elongational flow resists the breakup of the viscoelastic jet, leading to the formation of long threads of minijets. The splitting and splaying of the primary jet into a series of minijets is drastically boosted by the application of an electric field to the solution. If the non-Newtonian fluid is a solution, solvent evaporation from the minijets leads to a fibrous structure in the residual polymer [39].

Polymer concentration is another significant parameter, alongside molecular weight, that can alter the characteristics of the fibrous structures produced from the jet. Moreover, the concentration plays a major role in stabilizing the fibrous structure. Herein, the increase of the solution concentration resulted in a significant increase of the average diameter size of the fibers and the interfiber spacing. A similar pattern was observed previously [40,41], where the fiber diameter increased with increasing polymer concentration. This was attributed to a higher solid content resulting from a higher viscosity that would oppose the flow and elongation and result in a less-stretched jet [42].

The effect of the applied voltage on the fiber diameter has been a debated subject. It is commonly observed that higher voltages facilitate the formation of large diameter fibers [43].It has also been shown that by increasing the voltage of the electric field, the probability of beads formation increases. Nevertheless, a high electric field voltage may also result in multiple jets, which deliver a smaller diameter of electrospun fibers. Some studies have also reported the formation of thinner fibers upon boosting the electrical field due to the increased stretching of the electrospinning jet [44,45]. It is anticipated that a higher applied voltage produces fibers with small diameters, depending on the volatility and viscosity of each solvent. It is also noted that when the electric field is amplified during the electrospinning process, the charged jet travels much faster to the collector. Consequently, the solvent in the jet has less time to evaporate and, if the solvent has a lower vapor pressure, wet fibers are obtained with larger diameters. In the present work, the chloroform used as solvent has a very high volatility since it evaporates very fast at room temperature, and thus conforms nicely to the desired pattern [46]. The applied voltage may also affect some aspects such as the mass of polymer coming out from the tip of the needle, the elongation level of a jet by an electrical force, the morphology of a jet (a single or multiple jets), etc. Thus, we can conclude that a correct combination among these factors may define the desirable final diameter of electrospun fibers [22].

The flow rate of the polymer within the syringe is another important process parameter in electrospinning. A lower flow rate is more necessary as the solvent is given enough time to evaporate. There should always be a minimum flow rate of the spinning solution.In this work, the results indicated that the diameters of the electrospun PCL nanofibers increased, as the flow rate became lower. This is due to the fact that when decreasing the flow rate, the solvent evaporated, leading to the formation of solid nanofibers. Ideally, the feeding rate must match the solution removing rate from the tip. These findings suggest that in general, lower feeding rates can inhibit electrospinning and high feeding rates result in large diameter fibers due to the unavailability of solvent to evaporate within the time it takes to reach the collector [47].

The distance between the needle and collector can affect the fiber properties, particularly its diameter and morphology, as it affects the electrical field strength between the collector and tip of the needle. Specifically, the distance plays a significant part in the whipping, deposition time and evaporation rate of the solvent [45]. It has been reported that both lowering and increasing the voltage appear to reduce the fiber diameter, while an intermediate distance would fabricate the finest fiber. The effect of the voltage and the spinning distance in the current study was unexpected but reasonable. A high voltage could result in a smaller fiber diameter, when the distance is long enough to allow a more extension of the jet. On the other hand, a high voltage could result in a larger fiber diameter, if short spinning distance or high polymer concentration do not allow a substantial elongation of the jet [42]. 

PCL having Mn 71 kg/mol was selected for the incorporation of TFL in three different weight ratios (10, 20 and 30 wt%). Three different 17% *w*/*v* PCL/chloroform solutions were prepared and electrospun using the system parameters described in trial seven of Table 2 as setup conditions, since fibers with the largest mean diameter were obtained. These fibers were prepared and studied by various techniques, such as SEM, FT-IR, DSC and XRD, in order to investigate the morphology of the obtained fibrous structures, the interaction of the drug with the polymeric matrix and its physical state. SEM micrographs revealed an increase in the mean diameter of the obtained fibers with the incorporation of TFL as expected, with the values ranging from 0.492 to 0.653 μm. Drug/polymer interactions have a great effect on the rate of release of drugs from a polymer matrix. Polymers that can physically interact with the drug may sustain the release of the active substance. Drug–polymeric carrier interactions in the solid state are usually investigated using FT-IR technique by examining the wavelength shifts in the characteristic peak positions of either the drug or the polymer. Spectra regions where the peaks do not overlap are theoretically useful [48]. This technique is a powerful tool for investigating such systems and may detect changes in the vibrational frequencies of specific functional groups within the drug and polymers, due to H-bonds formation or other molecular interactions. Indeed, band shifting and/or broadening as well as band intensity variation of the relevant bands are markers of hydrogen bond formation [49].PCL evaluated in the present study consisted mainly of ester bonds and terminal carboxylic and hydroxyl groups, which can interact via hydrogen bonds, with the ester groups or amino groups of TFL and the nitrogen atoms in the TFL molecule. Therefore, to determine if there were such interactions in the prepared patches, we focused our analysis on the characteristic peaks recorded in the region of the hydroxyl and carbonyl groups of the FT-IR spectrum [50]. Noticeable peak shifts were observed mostly in the region of the ester group of PCL, suggesting intermolecular interactions (hydrogen bonds) between the C=O of PCL and -NH of the drug in all samples. Moreover, nitrile alkyl interactions [25] were also observed in all three samples in the area of 2220–2225 cm^−1^. Hence, in all cases, significant molecular interactions between the drug and the polyester were observed in all prepared nanofibers, independently of the TRL concentration.

DSC and XRD techniques were employed for the examination of the physical state of teriflunomide incorporated in the nanofibrous mats.It has been widely known that when poorly soluble drugs are prepared in their glassy, higher free-energy (amorphous) form, many poorly soluble drugs exhibit significantly higher solubility and faster dissolution than in their crystalline form.This advantage of enhanced dissolution rate and solubility can lead to enhanced bioavailability inside the patient’s organism and therefore improve the therapeutic result [51]. The melting point at 228.7 °C of neat TRL was not observed in the obtained DSC thermograms as seen in the thermograms of the fibrous patches, while only a peak was seen, corresponding to the melting of PCL at temperatures similar to those found in the pure fibrous structures, without the incorporation of TFL. This is an indication that TFL could be incorporated in amorphous form inside the nanofibers. However, due to the low melting point of PCL, TFL could have been dissolved in the melt of PCL during the DSC measurement (in situ melting). For this reason, the crystallinity of PCL and TFL in the fabricated fibers was also evaluated using the XRD method.Polymer crystallinity determines the polymeric degradability as well as the drug release, since the bulk crystalline phases are more unreachable to aquatic media [17]. The polymers studied in this work were semicrystalline and the fabricated fibers were anticipated to be semicrystalline too.XRD diffractograms showed the three characteristic peaks corresponding to the orthorhombic crystalline structure of pure PCL. Additional characteristic peaks corresponding to TFL were not noticed in the drug-loaded fibrous mats. This is attributed to the complete solubilization and amorphization of the drug within the polymeric matrix and is in accordance with the DSC thermographs analyzed previously.

The drug delivery of active molecules via electrospun fibrous mats has been examined extensively in the literature because the release behavior is mainly influenced in proportion to the structure of the formulation. When the resulting nanofiber mats are placed in aqueous media (e.g., human fluids), the system continuously delivers the drug, and meanwhile, the nanofibers are degraded [19]. The release rate of a drug from polymeric systems depends on many drug-related factors (such as its crystallinity and in general its physical state), but also on the polymeric carrier (molecular weight, melting point, etc.). Usually, depending on their characteristics, the molecules of the active substance that are close to the surface of the polymeric matrix are released more quickly. Over time, the hydration of the matrix helps the drug molecules of the inner core to be released (though diffusion) at a slower rate. Once released from the patch, the drug can be delivered to the joint cavity via two routes, i.e., direct diffusion at the site of application or through systemic circulation [52]. In vitro studies conducted herein revealed a biphasic release profile of TFL from the fibrous mats, consisting of a burst release within the first 10h as well as a sustained release up to 250 h. Alhusein et al. in their research also observed a similar release from tetracycline from zein/polycaprolactone electrospun matrices [53]. This may be related to the rapid dissolution of the active substance that is not incorporated inside the PCL matrix and is located at the surface of the fibers combined to the diffusion in the interfiber space (first phase), as well as to the hydrophobic nature, swelling and erosion properties of the polymer controlling the diffusion process of the drug in the fibers (second phase) [54,55,56]. Hydrophobicity plays a crucial role in drug delivery for extended and sustained release of active substances, as it lowers wettability of the polymeric matrix, therefore delaying the release of drugs [57].

After the in vitro release of teriflunomide from the fibrous patches, the SEM technique was used to determine the state of PCL. The main mechanisms through which a drug is released from a biodegradable polymer are swelling, diffusion and polymer erosion [58]. In general, the release is governed by the combination of all of these mechanisms, but it is mostly dependent on the relative rates of erosion and diffusion. Moreover, the porosity of the electrospun mats is another contributing aspect in controlling the release mechanism. Most biodegradable polymers used for drug administration are degraded. Synthetic polymers degrade by hydrolysis or biodegradation through cleavage of its backbone ester group hydrolysis to alcoholic and carboxylic end groups, leading to chain scission and the formation of oligomers and, finally, monomers. The degradation process for these polymers mostly affects the entire polymer matrix, leading to a uniform mode of erosion, called a “bulk” pathway. As water molecules break the chemical bonds along the polymer chain, the natural integrity of the polymer deteriorates allowing the drug to be released [59]. PCL employed in the designed patch is a slow-degrading polymer that enables a gradual degradation after implantation, thus avoiding extra costs and trauma related with secondary patch-removal procedures [28,29,30,56,60,61]. The observation of the SEM microphotographs of the fibrous mats after the in vitro release suggested that PCL/TFL samples exhibited three different drug release mechanisms: (i) the release of drug located at the surface of the fibers exposed to the outer water phase, (ii) the diffusion through the interfiber pores and finally (iii) the erosion of the polymeric matrix. Regarding the analysis of the release from PCL/TFL samples, it was noticed that the mass transfer resistance from the solid interface to the bulk liquid could be considered negligible. The first mechanism corresponds to diffusion and refers to the diffusion of the external drug through the interfiber voids. The diffusion occurs at a slab geometry (approximately the geometry of the patch). This mechanism can be associated with the first fast stage of the release process. The second mechanism refers to the diffusion of the TFL trapped within the fibers through the intrafiber pores to the surface of the fibers (from where it has to be diffused through the patch by the first mechanism). The third mechanism refers to a spontaneous release of the intrafiber TFL as the fiber suffers degradation. The direct modeling of the release process is extremely complicated. There are two diffusion problems in series and with changing geometry. According to the SEM data, the geometry of the problematic mechanism (ii) gradually changes from cylindrical to spherical. However, this is an oversimplification if the several processes occurring have different time scales. It is assumed that the mechanism (i) is much faster than mechanisms (ii) and (iii). In the absence of any quantitative information on PCL degradation, the relative contribution of mechanisms (ii) and (iii) on the release cannot be assessed. It is considered that the diffusion mechanism dominates. In this case the two processes can be assumed to act independently.

## 5. Conclusions

In this study, polycaprolactone polyesters of three different molecular weights were synthesized for the fabrication of nanofibers using the electrospinning process. According to the investigated parametric systems, the polymer’s concentration and molecular weight were found to play a significant role in controlling the morphology and diameter of the electrospun nanofibers while the voltage of the electric field, feed rate and distance from the collector were less effective compared to those parameters. Therefore, it can be noted that the morphology and diameter of the electrospun nanofibers were primarily affected by the molecular weight and polymer concentration in the electrospun solution (primary parameters), followed by the applied voltage field, flow rate and distance between the needle and the collector (secondary parameters). In addition, the obtained results indicated that the incorporation of TFL increased the fiber diameter of the PCL samples. Specifically, as the concentration of TFL increased, the mean fibers diameter was also increased in the nanofibrous mats. FT-IR measurements revealed several interactions between PCL and the drug. Nevertheless, the crystallinity of TFL was not observed in DSC and XRD tests, indicating amorphous dispersion. In vitro release studies showed a two-phase pattern, comprised by an initial burst release, followed by a sustained release of the drug up to 240 h. In conclusion, PCL nanofibrous mats loaded with TFL show promising results as candidates for drug delivery applications in the treatment of multiple sclerosis.

## Figures and Tables

**Figure 1 pharmaceutics-14-01693-f001:**
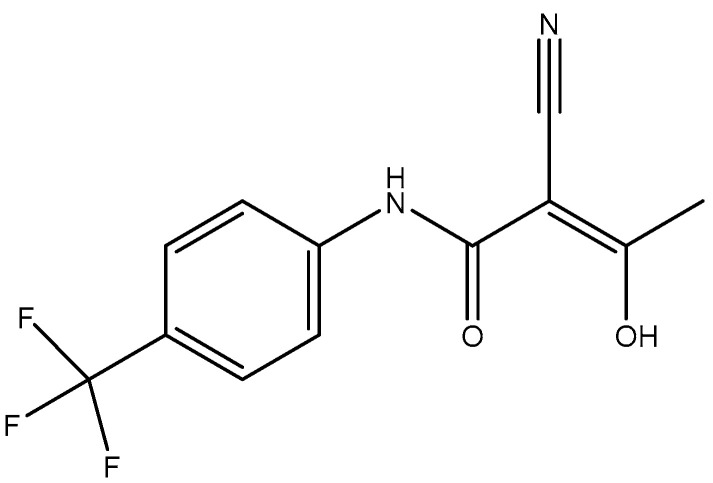
Chemical structure of teriflunomide (TFL).

**Figure 2 pharmaceutics-14-01693-f002:**
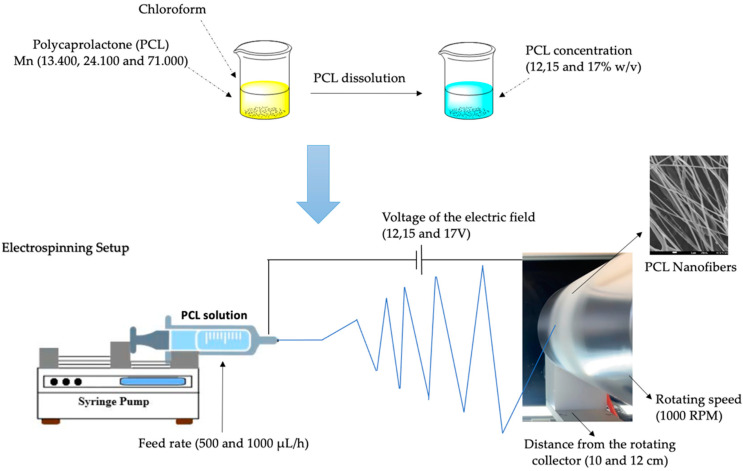
Setup of the electrospinning process and examined parameters.

**Figure 3 pharmaceutics-14-01693-f003:**
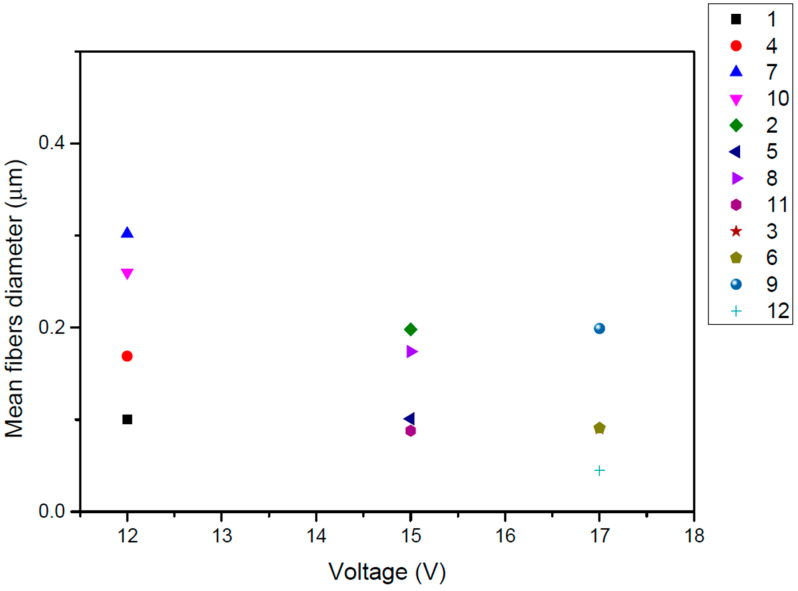
Effect of the applied voltage of the electric field on the fibers diameter (μm) for PCL (Mn 71.000) from 17% *w*/*v* chloroform solution for each investigated system of parameters.

**Figure 4 pharmaceutics-14-01693-f004:**
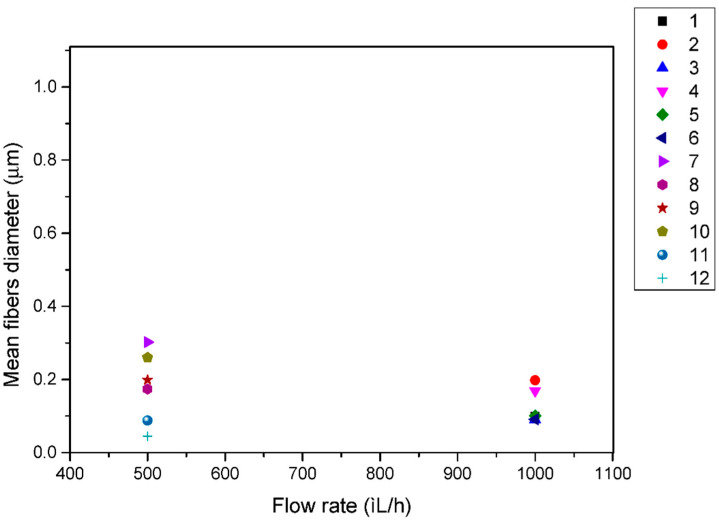
Effect of the flow rate on the fibers diameter (μm) for PCL (Mn 70,976) from 17% *w*/*v* chloroform solution for each investigated system of parameters.

**Figure 5 pharmaceutics-14-01693-f005:**
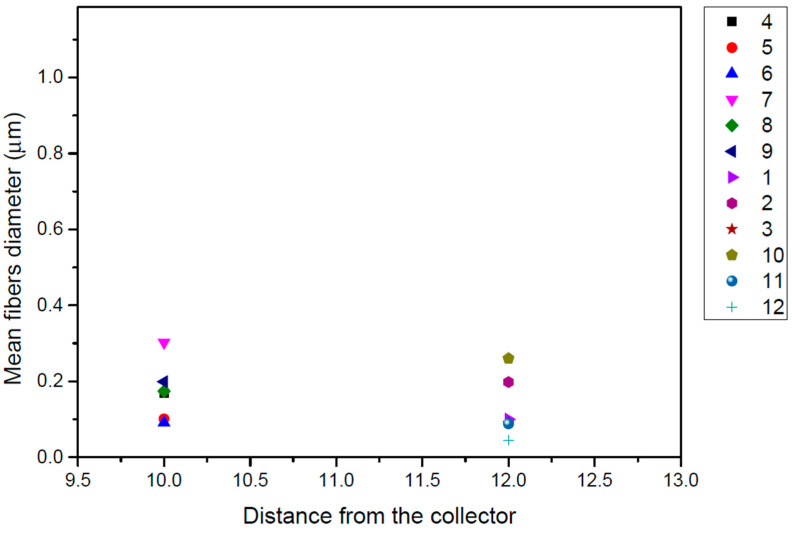
Effect of the distance between needle and the collector on the fibers diameter (μm) for PCL (Mn 71,000) from 17% *w*/*v* chloroform solution for each investigated system of parameters.

**Figure 6 pharmaceutics-14-01693-f006:**
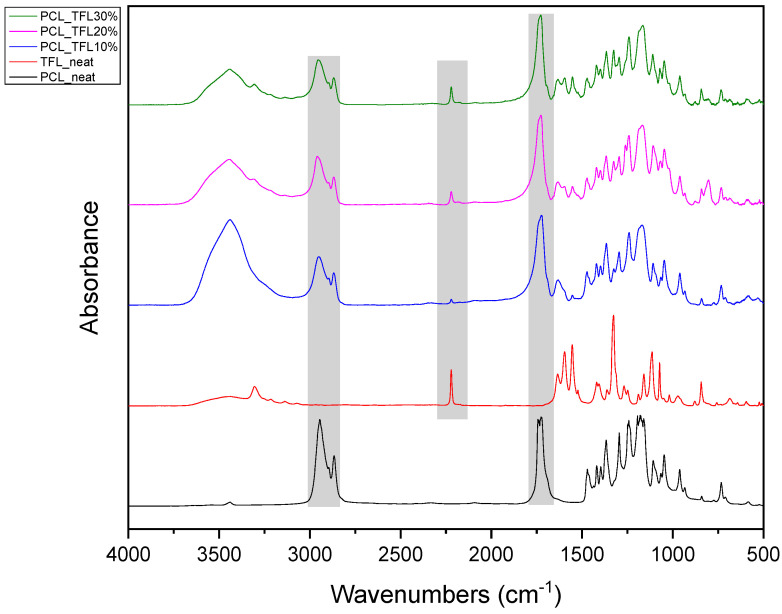
FT-IR spectra of neat PCL, neat teriflunomide and drug-loaded electrospun mats.

**Figure 7 pharmaceutics-14-01693-f007:**
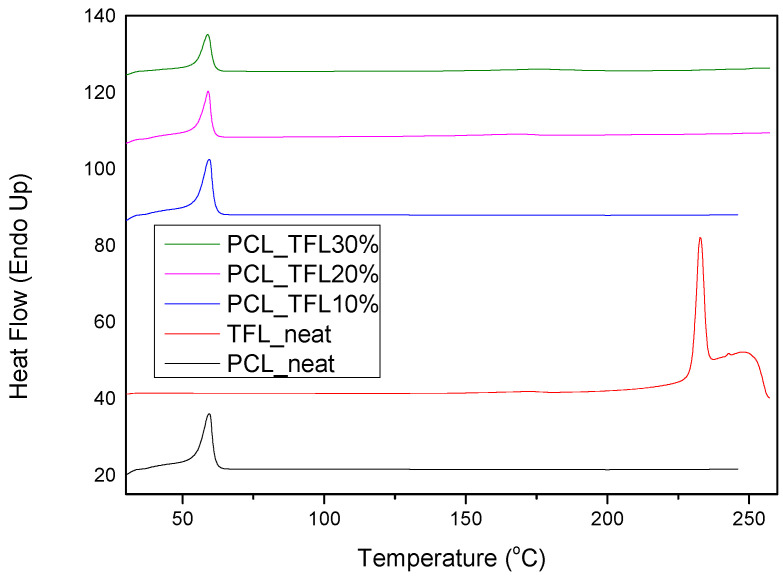
DSC the rmograms of PCL neat, teriflunomide neat and drug-loaded fibrous mats.

**Figure 8 pharmaceutics-14-01693-f008:**
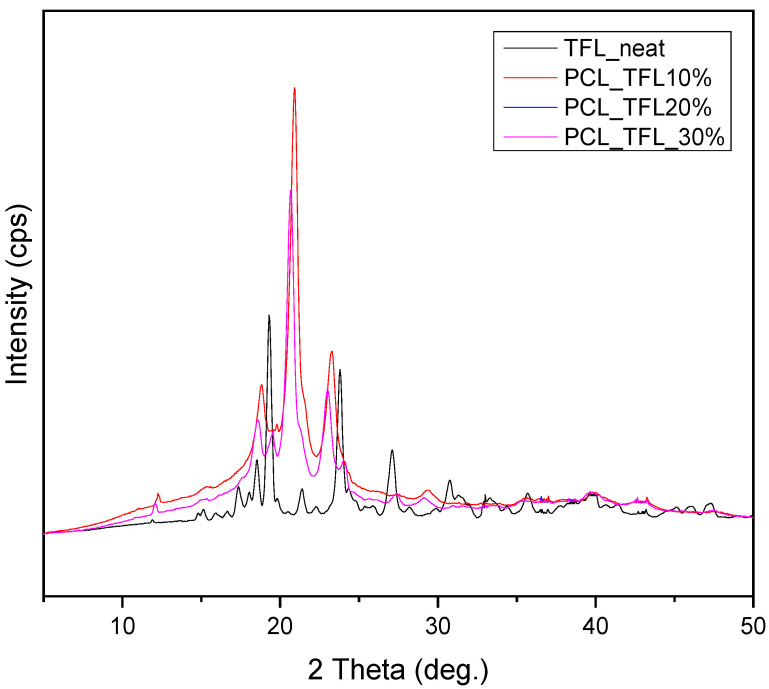
XRD diffractograms of the drug-loaded samples.

**Figure 9 pharmaceutics-14-01693-f009:**
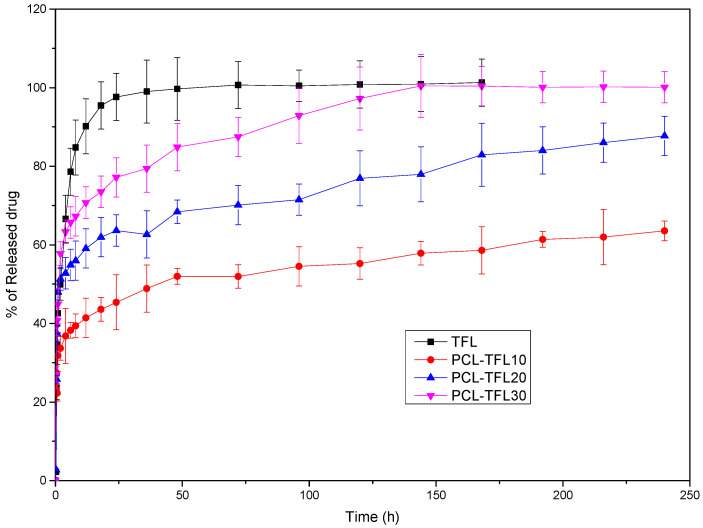
In vitro dissolution profiles of the prepared TFL loaded nanofibrous patches.

**Figure 10 pharmaceutics-14-01693-f010:**
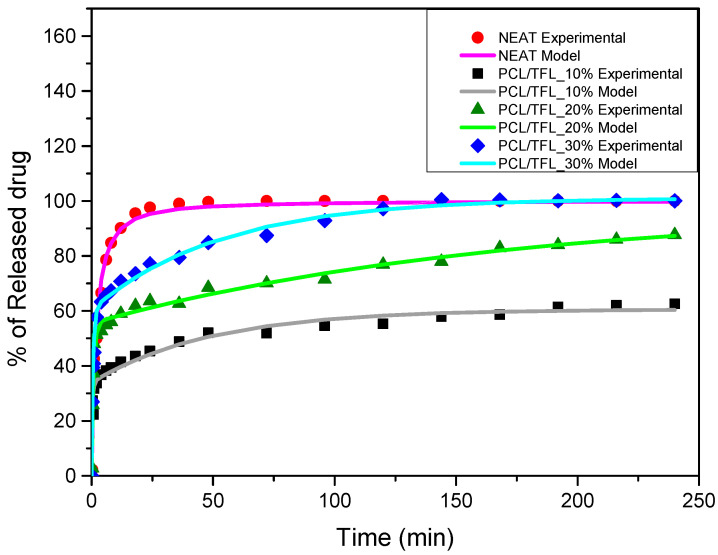
Comparison between the experimental (symbols) and model (continuous lines) release data for the neat TFL and TFL loaded nanofibrous patches.

**Table 1 pharmaceutics-14-01693-t001:** Different molar ratios of initiator and co-initiator used for the synthesis of PCL polyesters with different molecular weights.

[ε-CL]/[TEH (I)]	[ε-CL]/[1,4-Butanediol (CI)]	Mn (g/mol)
10,000	100	13.400
10,000	250	24.100
10,000	750	71.000

**Table 2 pharmaceutics-14-01693-t002:** Set of parameters of the electrospinning process for the preparation of the PCL electrospun nanofibrous matrices.

System of Parameters	Flow Rate (μL/h)	Rotating Speed RPM	Voltage (kV)	Distance (cm)
1	1000	1000	12	12
2	1000	1000	15	12
3	1000	1000	17	12
4	1000	1000	12	10
5	1000	1000	15	10
6	1000	1000	17	10
7	500	1000	12	10
8	500	1000	15	10
9	500	1000	17	10
10	500	1000	12	12
11	500	1000	15	12
12	500	1000	17	12

**Table 3 pharmaceutics-14-01693-t003:** SEM micrographs depicting the effect of the molecular weight of polymer on the formation of nanofibrous structures.

Sample	PCL 13,400	PCL 24,100	PCL 71,000
Fibers	Beads	Beads	Fibers’ formation
SEM images	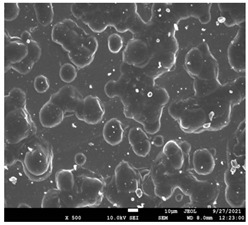	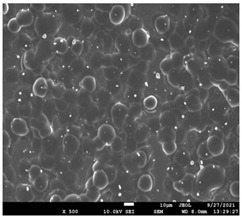	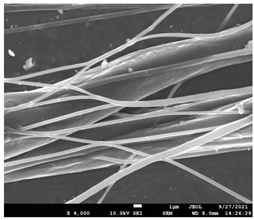

**Table 4 pharmaceutics-14-01693-t004:** SEM images presenting the effect of concentration of the polymer solution on the electrospun structure of PCL with 71,000 g/mol.

Concentration of PCL 71,000 g/mol in Chloroform (% *w/v*)	Mean Fibers Diameter (μm)/Standard Deviation	SEM Images
12	0.087/0.033	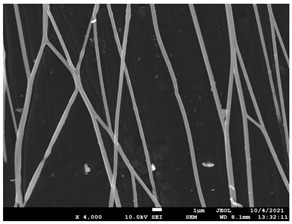
15	0.144/0.068	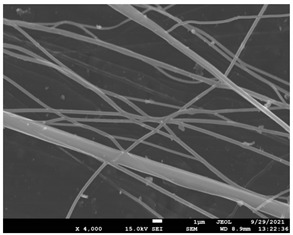
17	0.302/0.284	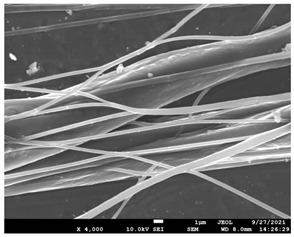

**Table 5 pharmaceutics-14-01693-t005:** Qualitative evaluation of electrospun samples.

Mn (g/mol)	Polymer Concentration (% *w/v*)	Qualitative Evaluation of Electrospinning	Optical Observation
13,400	12	No	No formation of fibers was observed, small beads obtained.
	15	No	No formation of fibers was observed, large beads obtained.
	17	No	No formation of fibers was observed, large beads obtained.
24,100	12	No	No formation of fibers was observed, small beads obtained.
	15	No	No formation of fibers was observed, small beads obtained.
	17	No	No formation of fibers was observed, large beads obtained.
71,000	12	Yes	The formation of continuous fibrous structure and low interfiber spacing was observed.
	15	Yes	The formation of continuous fibrous structure and low interfiber spacing was observed.
	17	Yes	The formation of continuous fibrous structure and high interfiber spacing was observed.

**Table 6 pharmaceutics-14-01693-t006:** SEM micrographs of PCL/TFL (10,20 and 30%) loaded nanofibers.

PCL_TFL10%	PCL_TFL20%	PCL_TFL30%
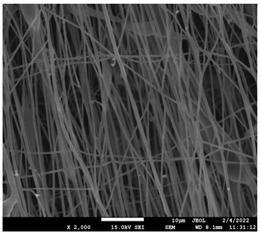	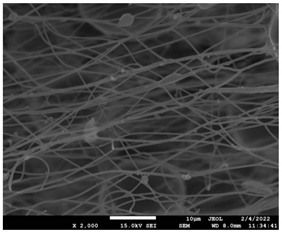	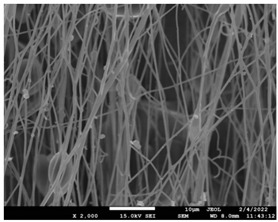
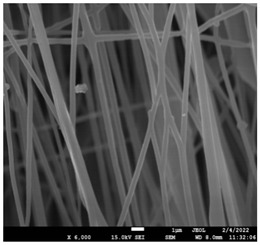	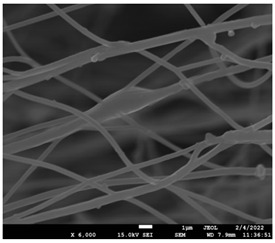	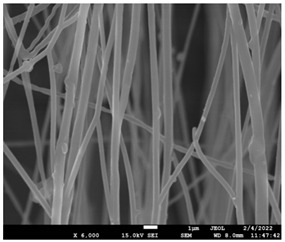
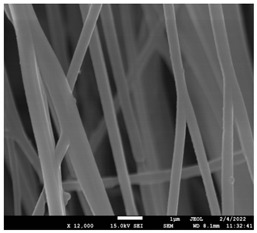	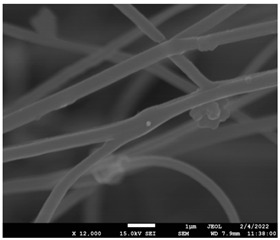	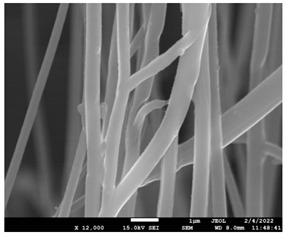

**Table 7 pharmaceutics-14-01693-t007:** Mean fibers’ diameter (μm), standard deviation, highest and lowest diameter (μm) value for drug-loaded fibers.

Samples	Mean Fibers Diameter (μm)	Standard Deviation	Highest Diameter (μm)	Lowest Diameter (μm)
PCL_TFL10%	0.492	0.188	0.841	0.464
PCL_TFL20%	0.541	0.223	0.946	0.352
PCL_TFL30%	0.653	0.135	1.119	0.387

**Table 8 pharmaceutics-14-01693-t008:** SEM images at different magnifications of PCL/TFL fibrous mats after in vitro drug dissolution for the investigation of release mechanism.

PCL_TFL10%		
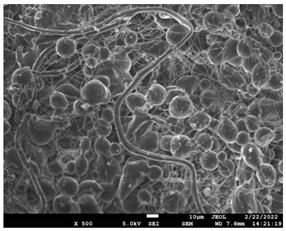	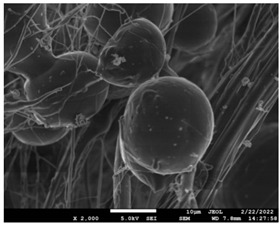	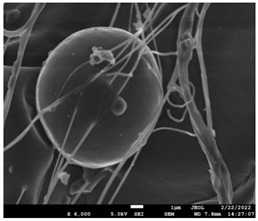
PCL_TFL20%		
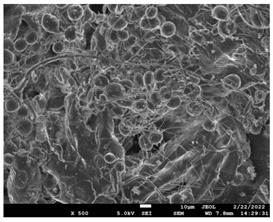	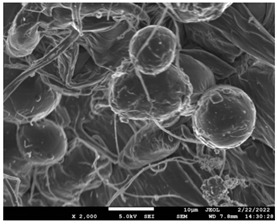	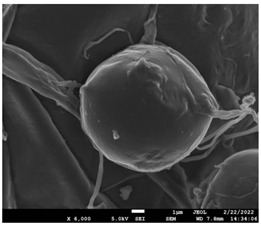
PCL_TFL30%		
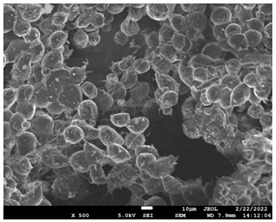	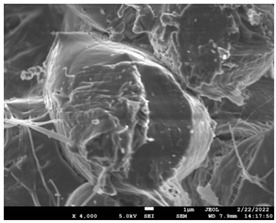	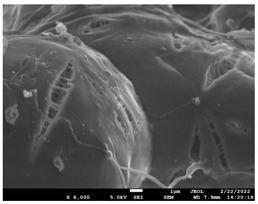

**Table 9 pharmaceutics-14-01693-t009:** Estimated parameter derived by fitting of the experimental release data.

	φ_1_	φ_2_	k_1_ (h^−1^)	k_2_ (h^−1^)
PCL/TFL 10%	0.34	0.27	2.965	0.02
PCL/TFL 20%	0.55	0.41	1.3	0.006
PCL/TFL 30%	0.60	0.41	1.71	0.019

## Data Availability

Not applicable.

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
