# Peer review of "Investigation of Molecular Weight, Polymer Concentration and Process Parameters Factors on the Sustained Release of the Anti-Multiple-Sclerosis Agent Teriflunomide from Poly(ε-caprolactone) Electrospun Nanofibrous Matrices"

_pharmaceutics, 2022, doi:10.3390/pharmaceutics14081693_

Round 1

Reviewer 1 Report

In this paper, the authors prepared some different PCL based patches via electrospinning for the sustained release of teriflunomide. They optimized the process conditions and characterized the different prepared matrices. The paper is well organized and can be published following minor revisions.

Please, don’t use the comma between the subject and the verb. See, for example, line 64 or line 65.

PCL based electrospun fibers have been used for a variety of applications which should be named in the introduction. See, for example, 10.3390/pr9122244, 10.3390/molecules27123739 and 10.1021/acsabm.2c00126.

Something went wrong in lines 148 and 149.

Eliminate the commas in line 226.

Table 4, Table 7: please use the point and not the comma for the decimal numbers.

Figure 3, Figure 4, Figure 5: please, correct the y axis. I imagine the unit is in micron and not in nanometers.

Reviewer 2 Report

1. Authors claim that (line 34) “significant molecular interactions were formed between the drug and the polyester”

Later in the manuscript (line 35) Authors also claim:

“Additionally, in vitro dissolution studies showed that the  PCL/TFL loaded nanofibers exhibit a biphasic release profile, having an initial burst release phase, followed by a sustained release until 250 h”

How would you explain this effect? (the burst effect)

Significant molecular interactions should inhibit the drug release. Shouldn’t?

2. There is a misunderstanding in lines: 316-329. Authors claim that:

“Pure TFL exhibits characteristic peaks at 3305 cm−1 and 3067 cm−1, which correspond to strong absorptions of hydroxyl (-OH) and secondary amino groups (-NH), at 1636 cm−1  , attributed to the carbonyl group (C=O) , and 2220 cm-1, corresponding to the triple nitrogen carbon bond trend in the nitrile (−C≡N) group. PCL is recorded at 1744 cm-1 and is due to the absorption of the carbonyl group of the ester 320 bonds.

Later, Authors write: “Looking at the obtained FT-IR spectra of the TFL loaded 322 nanofibers, it is obvious that when TFL was incorporated into the PCL polyester fibers, the absorption peak of the amino group at 1636 cm-1 was shifted”

Please explain.

3. The proof for the intermolecular hydrogen bonds formed between the C=O of PCL and -NH of TRL might be not only shifts of the signals but also e.g. the decrease of the intensity. Did Authors checked this?

 4. Line 133: “The collected electrospun samples were dried for 24 h, at ambient temperature to remove chloroform.”

According to the Reviewer’s experience, 24h is too short to remove the solvent from the nonwovens, did Authors check this?

Round 2

Reviewer 2 Report

The authors answered all doubts and I have no other objections.